# Pediatric *Candida* Bloodstream Infections Complicated with Mixed and Subsequent Bacteremia: The Clinical Characteristics and Impacts on Outcomes

**DOI:** 10.3390/jof8111155

**Published:** 2022-10-31

**Authors:** Wei-Ju Lee, Jen-Fu Hsu, Yu-Ning Chen, Shao-Hung Wang, Shih-Ming Chu, Hsuan-Rong Huang, Peng-Hong Yang, Ren-Huei Fu, Ming-Horng Tsai

**Affiliations:** 1College of Medicine, Chang Gung University, Taoyuan 333, Taiwan; 2Division of Pediatric Emergency Medicine, Department of Pediatrics, Chang Gung Memorial Hospital, Chiayi 618, Taiwan; 3Division of Pediatric Neonatology, Department of Pediatrics, Chang Gung Memorial Hospital, Taoyuan 333, Taiwan; 4Department of Microbiology Immunology and Biopharmaceuticals, National Chiayi University, Chiayi 618, Taiwan; 5Division of Neonatology and Pediatric Hematology/Oncology, Department of Pediatrics, Chang Gung Memorial Hospital, Yunlin 638, Taiwan

**Keywords:** candidemia, intensive care unit, mixed Candida/bacterial BSI, bloodstream infection, mortality

## Abstract

**Background:** Pediatricians face a therapeutic challenge when patients with *Candida* bloodstream infections (BSIs) simultaneously have positive bacterial culture. We aim to characterize the clinical characteristics of pediatric *Candida* BSIs complicated with mixed bacteremia and subsequent bacterial infections, risk factors and impacts on outcomes. **Methods:** All episodes of pediatric *Candida* BSIs between 2005 and 2020 from a medical center in Taiwan were reviewed. Mixed *Candida*/bacterial BSIs were defined as isolation of a bacterial pathogen from blood cultures obtained within 48 h before or after the onset of *Candida* BSI. The clinical features and impacts of mixed *Candida*/bacterial BSIs were investigated. **Results:** During the study period, 320 patients with a total of 365 episodes of *Candida* BSIs were identified and analyzed. Mixed *Candida*/bacterial BSIs were 35 episodes (9.6%). No significant difference was found between mixed *Candida*/bacterial BSIs and monomicrobial *Candida* BSIs in terms of patient demographics, *Candida* species distributions, most chronic comorbidities or risk factors. Patients with mixed *Candida*/bacterial BSIs were associated with a significantly higher risk of subsequent bacteremia (51.4% vs. 21.2%, *p* < 0.001) and a relatively higher candidemia-attributable mortality rate (37.2% vs. 22.4%, *p* = 0.061) than those with monomicrobial *Candida* BSIs. Mixed *Candida*/bacterial BSIs were not an independent risk factor of treatment failure or final mortality according to multivariate logistic regression analyses. **Conclusions:** The clinical significance of mixed *Candida*/bacterial BSIs in children included a longer duration of septic symptoms, significantly higher likelihood to have subsequent bacteremia, and relatively higher risk of candidemia attributable mortality.

## 1. Introduction

*Candida* bloodstream infections (BSIs) usually occur in critically ill patients or those with predisposing risk factors, such as long-term hospitalization in the intensive care unit (ICU), previous exposure to broad-spectrum antibiotics, the presence of artificial devices and underlying gastroenteropathy [1,2,3]. In children, *Candida* BSI is associated with high morbidity and mortality rates and reported in 28–46% of ICU patients or those with immunocompromised status [3,4,5]. Our previous studies found that underlying renal insufficiency, breakthrough candidemia, delayed catheter removal, and septic shock at onset were independently associated with final in-hospital mortality in children with candidemia [6,7]. Current studies have focused on the emergence of antifungal-resistant *Candida* species and the changing epidemiology from *C. albicans* to non-*albicans* candidemia after extensive antifungal prophylaxis [8,9,10].

Mixed *Candida*/bacterial bloodstream infections (mixed-BSIs) account for 18–46% of all *Candida* BSIs [11,12]. Although the mortality rate of mixed-BSIs is similar to that of monomicrobial *Candida* BSI, the clinical manifestations of mixed-BSIs seem more complicated [12,13,14] and a prolonged hospital course and significantly higher rate of infectious complications have been reported [12,13,14]. Pediatric patients with *Candida* BSIs remain at risk of other nosocomial infections, and some of them have subsequent recurrent candidemia or bacterial infections [8,15]. Clinicians face therapeutic challenges when patients with candidemia have sudden and unexpected clinical deterioration while they are still on antifungal therapy. They have to decide whether this is an episode of breakthrough candidemia or a new episode of infection [15,16,17]. Additionally, the issue of mixed *Candida*/bacterial BSIs has not been well investigated in pediatric patients with candidemia. In this study, we aim to characterize candidemia complicated with mixed and subsequent bacterial infections in children and investigate the influences of therapeutic strategies on outcomes.

## 2. Patients and Methods

### 2.1. Study Design, Setting and Ethics Approval

All pediatric patients with candidemia aged less than 18 years old and admitted to Chang Gung Memorial Hospital (CGMH) between January 2005 and December 2020 were enrolled and reviewed. The Linkou CGMH is the largest, university-affiliated teaching hospital in Taiwan, and the pediatric department of Linkou CGMH has 550 beds. The pediatric ICU and three NICUs have a total of 20 beds and 107 beds, respectively. Additionally, there is a specialized pediatric hematology/oncology ward on the sixth floor. In our institute, all blood cultures are processed using the BACTEC 9240 (Becton Dickinson Microbiology Systems, Franklin Lakes, NJ, USA). All the samples that have been sent to the laboratory are taken in a standard manner in terms of sampling criteria. During the study period, all *Candida* isolates of pediatric candidemia were retrieved from the central laboratory and re-identified using Matrix-assisted laser desorption ionization time-of-flight mass spectrometry (MALDI-TOF, Bruker Biotype, software version 3.0, USA). This study was approved by the institutional review board of the CGMH (certificate number: 202201214B0) and a waiver of informed consent was also approved for anonymous data collection and the retrospective design of this study.

### 2.2. Definitions and Data Collection

An episode of Candida BSI was defined when a patient had compatible signs or symptoms of BSIs and ≥1 positive blood culture for *Candida* species. For unidentified *Candida* spp. in the blood culture, the episode of *Candida* BSI was excluded. All episodes of candidemia and subsequent nosocomial infections after onset of *Candida* BSI were reviewed and analyzed. The onset of *Candida* BSI was defined as the first positive blood culture of the *Candida* isolates with compatible clinical symptoms and signs. Mixed *Candida*/bacterial BSIs were defined as the isolation of a bacterial pathogen from blood cultures obtained within 48 h before or after the onset of *Candida* BSI. In our institute, the blood cultures of patients with *Candida* BSI were usually repeated every 2 to 3 days until negative or when clinically indicated. When a bacterial pathogen was identified in the blood culture at 48 h after onset of *Candida* BSI, until the patient had completed the antifungal therapy or death, we considered it as *Candida* BSI complicated with subsequent bacteremia.

Blood cultures positive for the following microorganisms were considered to be contaminants, including Corynebacterium, Propionibacterium, micrococcus species, and Bacillus species, and were excluded. The diagnosis of neonatal sepsis, including sepsis due to coagulase negative staphylococcus (CoNS), was based on the standard criteria applied by the previous publications [18,19]. The subsequent episode of *Candida* BSI was considered independent from the first episode when the patient had completed the antifungal therapy, two or more negative blood cultures from the last positive culture growing a *Candida* isolate, and resolution of all clinical symptoms of candidemia [8,15]. Breakthrough candidemia was defined as a new onset of *Candida* BSI while this patient was still on antifungal therapy or antifungal prophylaxis [15,16,17].

Medical records were reviewed to determine the response to antifungal therapy at two weeks after the onset of *Candida* BSI based on the following guidelines for assessing treatment responses published by the Mycoses Study Group and European Organization for Research and Treatment of Cancer: complete response was resolution of candidemia and clinical symptoms within three days, partial response was within 7 days, and progression of disease and death were considered as “treatment failure” [20]. The demographic data, chronic comorbidities, medical courses and other risk factors associated with *Candida* BSI were also reviewed within 30 days before the positive blood cultures of *Candida* species. *Candida* BSI-attributable mortality was defined as (1) death before the resolution of signs and symptoms related to candidemia, or (2) death within 14 days after the onset of candidemia without other explanation. In subsequent bacteremia following candidemia, we considered *Candida* BSI-attributable mortality according to our previous studies [20].

### 2.3. Antifungal Susceptibility Testings

In vitro antifungal susceptibilities of all Candida isolates were performed based on the standard protocol, as described in our previous studies [7], according to the EUCAST-Antifungal Susceptibility Testing microdilution method. The Candida krusei* ATCC^®^ 6258 and Candida parapsilosis ATCC^®^ 22,019 were used as the quality control strains for antifungal drug susceptibility testing.

### 2.4. Statistical Analysis

All episodes of pediatric *Candida* BSIs were categorized as monomicrobial *Candida* BSI and mixed *Candida*/bacterial BSIs. The clinical characteristics, treatment and outcomes were compared between the case (mixed *Candida*/bacterial BSIs) and control groups. The demographic, clinical, outcome variables and in vitro susceptibility data were summarized using the descriptive statistics. Categorical variables were compared using the χ^2^ or Fisher’s exact test, and continuous variables by the Mann-Whitney *U* test. A *p*-value of 0.05 was considered significant.

The clinical significances and impacts of mixed *Candida*/bacterial BSIs were investigated and independent risk factors of candidemia-attributable mortality were evaluated. A univariate logistic regression was fitted for each variable to test its relationship with mortality outcomes. Variables clinically relevant and statistically significant (*p* < 0.1) on univariate analysis were considered to build the multivariate regression model. Clinical interventions were maintained in the final model as a fixed variable. All statistical analyses were performed using the IBM SPSS software (version 22.0; IBM SPSS Inc., New York, NY, USA).

## 3. Results

During the study period, a total of 320 patients with 356 episodes of *Candida* BSIs were identified. A total of 108 neonates with 123 episodes of candidemia and 212 pediatric patients with 242 episodes of candidemia participated. The patient demographics and underlying chronic comorbidities are summarized in Table 1. At the time of candidemia, 131 episodes (35.9%) occurred in the pediatric intensive care unit (PICU), 123 (33.7%) in the neonatal ICU (NICU), 91 (24.9) in pediatric wards and 20 (5.5%) in burn-surgical ICUs. Among them, a bacterial pathogen was isolated within two days before or after onset of positive *Candida* species in 35 episodes (9.6%) and categorized as the mixed *Candida*/bacterial BSIs. In total, candidemia was identified in 61 (16.7%) episodes, with *Candida* isolates recovered from intra-abdominal space or abscess (*n* = 37), pleural fluid (*n* = 9), urinary source (*n* = 9), and cerebrospinal fluids (*n* = 6). A total of 17 episodes of disseminated candidemia were noted in the cohort, which indicated that the *Candida* isolates were identified from more than two sterile sites. The distributions of *Candida* species were comparable between neonatal episodes and the pediatric patients with *Candida* BSIs (Table 1).

### 3.1. Microbiological Characteristics and Clinical Features

The isolated bacterial pathogens were shown in Figure 1. A total of 42 bacterial microorganisms other than *Candida* species were isolated from 35 episodes of mixed *Candida*/bacterial BSIs. The most common co-pathogen was Gram-negative bacteria (*n* = 25, 59.5%), followed by Gram-positive bacteria (*n* = 17, 40.5%). In terms of specific microorganisms, the most frequent pathogen was *Staphylococcus aureus* (*n* = 8, 19.0%), followed by *Klebsiella pneumonia* (*n* = 6, 14.3%) and coagulase-negative *Staphylococcus* (*n* = 5, 11.9%). A total of 7 episodes of mixed *Candida*/bacterial BSIs were caused by more than 2 microorganisms.

*C. albicans* accounted for more than half of all mixed *Candida*/bacterial BSIs (54.3%, *n* = 19). The clinical presentations including sources of candidemia, severity of illness, and underlying risk factors were comparable between monomicrobial *Candida* BSIs and mixed *Candida*/bacterial BSIs (Table 2). Severe sepsis occurred in 54.3% of mixed *Candida*/bacterial BSIs and 39.4% of monomicrobial BSIs (*p* = 0.104), whereas the occurrence rate of septic shock in patients with mixed *Candida*/bacterial BSIs was relatively higher than those with monomicrobial *Candida* BSIs (42.9 vs. 27.3%, *p* = 0.075).

Antifungal therapy was initiated after a median of 2 days (range, 0–7) after the first positive blood culture of *Candida* isolates. The median duration of antifungal treatments in the cohort was 17 (range, 1–68) days. In 11 cases (3.0%), the patient died before any antifungal agents were prescribed. Among the 354 episodes treated with antifungal agents, 157 episodes (44.4%) had modifications to the antifungal regimens during the treatment course. Notably, mixed *Candida*/bacterial BSIs were significantly more likely to be treated with azole agents than monomicrobial *Candida* BSIs (57.1 vs. 37.5%, *p* = 0.014). The percentage of delayed appropriate antifungal agents were comparable between the two groups. Although the percentage of antifungal agent modifications were comparable between these two groups, a significantly longer treatment duration was noted in patients with mixed *Candida*/bacterial BSIs than those with monomicrobial *Candida* BSIs. Patients with mixed *Candida*/bacterial BSIs were more likely to have a longer duration of septic symptoms, significantly higher risk of subsequent bacteremia (51.4% vs. 21.2%, *p* < 0.001) and relatively higher *Candida* BSI-attributable mortality rate (37.2% vs. 22.4%, *p* = 0.061) than those with monomicrobial *Candida* BSIs (Table 3). The median time between onset of subsequent bacterial infections and an episode of *Candida* BSI was 9 days (range: 3–35 days).

### 3.2. Antifungal Susceptibility Testing Results, Therapeutic Outcomes and Independent Risk Factor of Mortality

The antifungal susceptibilities have been performed for all these *Candida* isolates [7,8,15]. Although the standard minimum inhibitory concentration (MIC) cut-off point of resistance can be found for some uncommon *Candida* species, 85.3% of all *Candida* isolates were susceptible to fluconazole (MIC < 4 mg/L). Specifically, only 5 (2.9%) *C. albicans* isolates were resistant to fluconazole, but all *C. glabrata* isolates and 38.1% of *C. tropicalis* isolates were intermediate or resistant to fluconazole. Additionally, non-*albicans Candida* isolates had a significantly higher rate of azole resistance than *C. albicans* isolates. All *Candida* isolates were susceptible to amphotericin B, and resistance to anidulafungin, caspofungin and micafungin was uncommon.

Overall, the *Candida* BSI-attributable mortality rate was 21.1% (77 of 365) and the overall in-hospital mortality rate was 33.4% (107 of 320 patients died). The final treatment outcomes were comparable between patients with monomicrobial *Candida* BSIs and those with mixed *Candida*/bacterial BSIs. The treatment outcomes did not significantly change during the study period, when cases were divided into four study periods (2005–2008, 2009–2012, 2013–2016 and 2017–2020) and compared, although new antifungal agents, echinocandins, were more widely used since 2010 in our institute.

We tried to find the independent risk factors of candidemia-attributable mortality (Table 4). Neonates had significantly higher rates of sepsis-attributable mortality and in-hospital mortality than children (32.4% vs. 19.8%, *p* = 0.012 and 41.7% vs. 29.3%, *p* = 0.041, respectively). Delayed appropriate antifungal treatments (>48 h after onset of *Candida* BSIs), delayed catheter removal, breakthrough candidemia, more chronic comorbidities and septic shock at onset were associated with an increased risk of candidemia attributable mortality. After multivariate logistic regression analyses, the independent risk factors for candidemia attributable mortality in pediatric patients with *Candida* BSIs were breakthrough candidemia (OR, 4.59; 95% CI: 1.92–10.98, *p* = 0.002), delayed catheter removal (OR, 2.83; 95% CI: 1.39–5.74, *p* = 0.004), and septic shock at onset (OR, 4.65; 95% CI: 2.24–9.64, *p* < 0.001).

## 4. Discussion

To our knowledge, this is the first study to investigate the clinical significance of *Candida* BSIs complicated with mixed and subsequent bacteremia in children. Although *Candida* BSIs complicated bacterial infections did not significantly contribute to worse outcomes, we found that pediatric patients with mixed *Candida*/bacterial BSIs were more likely to have subsequent bacteremia and had more serious clinical presentations, including relatively higher rates of severe sepsis and septic shock. In children with candidemia, *Candida* isolates with antifungal resistance to azole agents accounted for only 14% of all episodes [7], and delayed catheter removal did have an impact on treatment failure. Although we cannot find the risk factors of mixed *Candida*/bacterial BSIs, the “sporadic” cases would cause therapeutic challenges in clinical practices and deserve more concern.

Bacterial pathogens are associated with *Candida* BSIs in approximately 25–56% of adult patients with candidemia [11,12,13,14,21,22]. Previous studies found that mixed *Candida*/bacterial BSIs had a lower clearance rate of candidemia, were more likely to progress to septic shock and multi-organ failure, and involved a longer duration of hospitalization and mechanical ventilation for patients [11,12,13,14]. However, very few studies were conducted in pediatric patients or neonatal ICU [21]. We found that mixed *Candida*/bacterial BSIs in children were relatively fewer than those in adults, accounting for only 9.6%, but *Candida* BSIs complicated with subsequent bacteremia were much more common than we had thought. In contrast to previous studies that found the presence of artificial devices and underlying chronic comorbidities to be associated with the occurrence of mixed *Candida*/bacterial BSIs [11,12,13,14], we could not find independent risk factors of mixed *Candida*/bacterial BSIs because almost all clinical and demographic characteristics were comparable between the monomicrobial *Candida* BSIs and mixed *Candida*/bacterial BSIs. Our data suggested that mixed *Candida*/bacterial BSIs were likely to be sporadic, similar to that found in polymicrobial BSIs of previous studies [23,24].

Because the duration of hospital and ICU stay are affected by underlying chronic comorbidities and patient demographics such as gestational age in our cohort, we cannot have similar findings to those concluded in previous studies [13,14]. However, we did find that patients with mixed *Candida*/bacterial BSIs had a significantly higher risk of subsequent bacteremia than those with monomicrobial *Candida* BSIs, which indicated that they would have longer hospital stay and use more resources. Previous studies also have found that recurrent bacteremia or candidemia were significantly associated with final worse outcomes [8,25], which is compatible with our study that patients with mixed *Candida*/bacterial BSIs would have a longer and more complicated hospital course. Therefore, cases of mixed *Candida*/bacterial BSIs deserve more attention and aggressive follow up for possible next episodes of infections is warranted.

While some studies have found significant worse outcomes in mixed *Candida*/bacterial BSIs [14,26,27], others discovered that the mortality rate was not significantly different [11,12,13]. The controversial results may result from different study cohorts, and different underlying demographics and study designs [11,12,13,14,26,27]. In our cohort, we noted the relatively higher candidemia-attributable mortality rate among episodes of mixed *Candida*/bacterial BSIs, which almost reached statistical significance, but it is not the independent risk factor of final in-hospital mortality. It is more likely that mixed *Candida*/bacterial BSIs would increase the risk of subsequent more nosocomial infections, which were independently associated with final adverse outcomes [8,25]. Based on our results and previous studies, we strongly recommend aggressive therapeutic strategies and early removal of an intravenous catheter [7,15,28], especially in patients with septic shock and multiple chronic comorbidities, to improve treatment outcomes.

The most common three bacterial pathogens of mixed *Candida*/bacterial BSIs were *Staphylococcus epidermidis*, *Klebsiella* spp. and *S. aureus* [11,12,13,14]. This may be explained by the higher likelihood of mixed biofilm formation [29]. In our cohort, 54.8% of bacterial pathogens were *Staphylococcus* spp. or *Klebsiella* spp., which were compatible with previous studies [11,12,13,14]. Several in vivo and in vitro models have explained the more severe clinical manifestations in coinfections involving the *Candida* and *Staphylococcus* species [26,29,30,31], which is compatible with the relative higher severity of illness and occurrence of septic shock in mixed *Candida*/bacterial BSIs in our cohort and previous studies [27,32]. Although new antifungal agents, echinocandin or caspofungin, have been launched in our institute since nearly one decade ago, this anti-biofilm agent did not significantly improve the outcomes. Therefore, higher attention toward specific cases with high risk of adverse outcomes and new therapeutic strategies are warranted in the future.

There were some limitations in this study. This is a single center, retrospective study, and there was inevitably some missing data or unidentified cases during the long study period. Our conclusion may be less applicable to general institutes, because the therapeutic policies and local epidemiology may be different. Because two populations were enrolled over a long period of nearly two decades, the therapeutic strategies may have been changed and affect the outcomes. The therapeutic duration and strategies, including modification of antifungal regimens and removal of catheter depended on the decisions of attending physicians, which may affect the outcomes Additionally, more subgroup analyses such as antifungal-resistant *Candida* species, formation of biofilm, and specific non-*albicans Candida* species were not performed in this study. Finally, the follow-up blood culture in our cases of *Candida* BSIs was not universal; therefore, some bias in the duration of candidemia may exist in this study.

In conclusion, in pediatric patients, mixed *Candida*/bacterial BSIs or *Candida* BSIs complicated with subsequent bacteremia were not uncommon. The mortality rate of *Candida* BSIs in children was high, at least partially due to the superimposed bacterial infections, especially in patients with underlying chronic comorbidities. Pediatric *Candida* BSIs complicated with bacterial infections were more likely to have a longer and complicated duration of hospital course, although mixed *Candida*/bacterial BSIs were not independently associated with worse final outcomes. New antifungal agents, including the anti-biofilm agent caspofungin, have not significantly improve the treatment outcomes in recent years. Since early removal of the intravenous catheter is especially important to avoid subsequent bacteremia and significantly affects final outcomes, this strategy should be emphasized. In cases of multiple chronic comorbidities or septic shock, more aggressive therapeutic strategies should be enforced.

## Figures and Tables

**Figure 1 jof-08-01155-f001:**
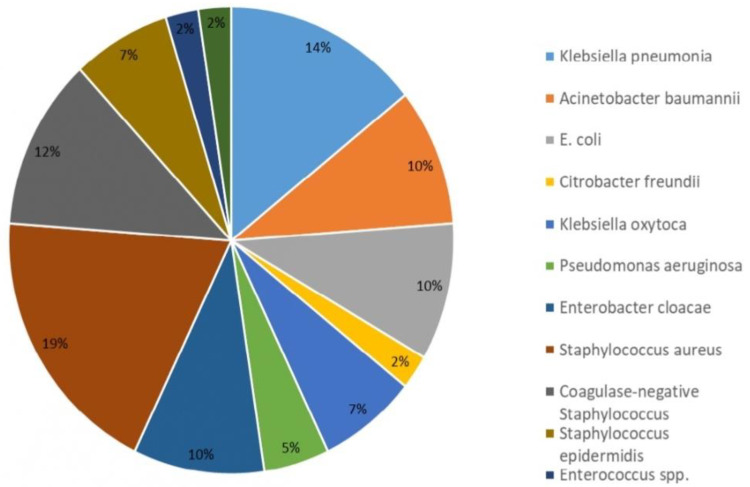
Distribution of the isolated bacterial pathogens in mixed Candida/bacterial bloodstream infections in the Department of Pediatrics, Chang Gung Memorial Hospital, 2005–2020.

**Table 1 jof-08-01155-t001:** Patient demographics of 320 children with *Candida* bloodstream infections in CGMH, 2005–2020.

Patient Characteristics (Total *n* = 320 Patients)	Neonates Age < 3 Months (*n* = 108 patients)	Pediatric Patients (*n* = 212 Patients)	*p* Values
**Patient demographics**			
Birth body weight (g), median (IQR)	1160.0 (762.0–2070)	-	-
Gestational age (weeks), median (IQR)	29.0 (26.0–35.0)	-	-
Age (year old), median (IQR)	-	4.5 (1.3–13.0)	-
Gender (male/female)	65 (60.2)/43 (39.8)	102 (48.1)/110 (51.9)	0.091
NSD/Cesarean section	52 (48.1)/56 (51.9)	-	-
Inborn/outborn, n (%)	80 (74.1)/28 (25.9)	-	-
5 min Apgar score ≤ 7, n (%)	44 (40.7)	-	-
Perinatal asphyxia, n (%)	7 (6.5)	-	-
Respiratory distress syndrome (≥Gr II), n (%)	57 (52.8)	-	-
Intraventricular hemorrhage (≥Stage II), n (%)	23 (21.3)	-	-
Day of life at onset of candidemia (day), median (IQR)	26.0 (14.0–62.0)	-	-
Duration of hospitalization before onset of candidemia (day), median (IQR)	-	29.5 (13.0–48.0)	-
**Underlying chronic comorbidities, *n* (%)**			-
Neurological sequelae	28 (25.9)	82 (38.6)	
Bronchopulmonary dysplasia or chronic lung disease	65 (60.2)	47 (22.2)	-
Complicated cardiovascular diseases	5 (4.6)	19 (9.0)	-
Symptomatic patent ductus arteriosus	25 (23.1)	0 (0)	-
Gastrointestinal sequelae	28 (25.9)	66 (31.1)	-
Renal disorders	7 (6.5)	38 (17.9)	-
Hematological/Oncology	0 (0)	46 (21.7)	
Immunodeficiency	0 (0)	6 (2.8)	
Autoimmune disorders	0 (0)	8 (3.8)	
Congenital anomalies	16 (14.8)	26 (12.3)	-
Presences of any chronic comorbidities	92 (85.2)	204 (96.2)	-
Presences of more than one comorbidity	43 (39.8)	107 (53.0)	-
**Case years, *n* (%)**			-
2005–2008	26 (24.0)	52 (24.5)	
2009–2012	28 (25.9)	55 (25.9)	-
2013–2016	30 (27.8)	56 (26.4)	-
2017–2020	24 (22.2)	49 (23.1)	-
**Candidemia-attributable mortality, *n* (%)**	35 (32.4)	42 (19.8)	0.012
**Overall final in-hospital mortality, *n* (%)**	45 (41.7)	62 (29.3)	0.041

**Table 2 jof-08-01155-t002:** Comparisons of monomicrobial *Candida* bloodstream infections (BSIs) and mixed *Candida*/bacterial BSIs in CGMH, 2005–2020.

	Overall *Candida* BSI (total *n* = 365)	Monomicrobial *Candida* BSIs (Total *n* = 330)	Mixed *Candida*/Bacterial BSIs (Total *n* = 35)	*p* Value
Neonatal episodes	114 (31.2)	106 (32.1)	8 (22.9)	0.338
Pediatric episodes	251 (68.8)	224 (67.9)	27 (77.1)	
Pathogens				0.708
*Candida albicans*	171 (46.8)	152 (46.1)	19 (54.3)	
*Candida parapsilosis*	95 (26.0)	84 (25.5)	11 (31.4)	
*Candida tropicalis*	21 (5.8)	21 (6.4)	0 (0)	
*Candida glabrata*	20 (5.5)	18 (5.5)	2 (5.7)	
*Candida guilliermondii*	18 (4.9)	18 (5.5)	0 (0)	
Other *Candida* spp.	40 (11.0)	37 (11.2)	3 (8.6)	
Source of candidemia *				0.602
Primary bloodstream infection (BSI)	226 (61.9)	200 (60.1)	26 (74.3)	
Catheter-related BSI	78 (21.4)	72 (21.8)	6 (17.1)	
Abdominal	37 (10.1)	35 (10.6)	2 (5.7)	
Urological	9 (2.5)	9 (2.7)	0 (0)	
Pulmonary	9 (2.5)	8 (2.4)	1 (2.9)	
Meningitis	6 (1.6)	6 (1.8)	0 (0)	
Clinical presentation				
Sepsis	301 (82.5)	272 (82.4)	29 (82.9)	0.982
Severe sepsis	149 (40.8)	130 (39.4)	19 (54.3)	0.104
Septic shock	105 (28.8)	90 (27.3)	15 (42.9)	0.075
Progressive and deteriorated ^¶^	72 (19.7)	62 (18.8)	10 (28.6)	0.181
Disseminated candidiasis ^$^	17 (4.7)	15 (4.5)	2 (5.7)	0.389
Duration of candidemia (days), median (range)	4.0 (1.0–42.0)	4.0 (1.0–36.0)	4.0 (1.0–42.0)	0.214
Predisposing risk factors ^#^				
Receipt of systemic antibiotics ^&^	340 (93.2)	305 (92.4)	35 (100.0)	0.151
Previous azole exposure ^&^	40 (11.0)	36 (10.9)	4 (11.4)	0.925
Previous bacteremia ^&^	184 (50.4)	163 (49.4)	21 (60.0)	0.287
Presence of CVC	354 (97.0)	321 (97.3)	33 (94.3)	0.285
Receipt of parenteral nutrition	250 (68.5)	224 (67.9)	26 (74.3)	0.566
Receipt of immunosuppressants	70 (19.2)	64 (19.1)	6 (17.1)	0.748
Artificial device other than CVC	180 (49.3)	163 (49.4)	17 (48.6)	0.926
Prior surgery ^&^	111 (30.4)	105 (31.8)	6 (17.1)	0.117
Neutropenia (ANC < 0.5 × 10^3^/μL)	87 (23.8)	76 (23.0)	11 (31.4)	0.297

All data are expressed as case number (%), unless otherwise stated. * Source of candidemia was defined as the first sterile site to have positive culture for the *Candida* species in the episode. ^¶^ Defined as candidemia episodes with more disseminated candidiasis and/or progressive multi-organ failure even after effective antifungal agents. ^#^ Indicateds the presence of underlying condition or risk factor at onset of *Candida* BSI, and most episodes occurred in patients with >1 underlying conditions or risk factors. ^&^ Within one month prior to onset of invasive candidemia. ^$^ Indicates positive *Candida* isolates recovered from more than two sterile sites, in addition to primary bloodstream infection.

**Table 3 jof-08-01155-t003:** Therapeutic strategies and outcome comparisons of pediatric Candida bloodstream infections (BSIs) and mixed Candida/bacterial BSI.

Variable	Overall *Candida* BSIs (Total *n* = 365)	Monomicrobial *Candida* Bloodstream Infections (Total *n* = 330)	Mixed *Candida*/Bacterial Bloodstream Infections (Total *n* = 35)	*p* Value
Final treatment regimens				0.041
Fluconazole/Voriconazole	137 (37.5)	117 (35.5)	20 (57.1)	0.016
Amphotericin B	106 (29.0)	107 (32.4)	5 (14.3)	0.050
Echinocandin-based regimen	103 (28.2)	96 (29.1)	7 (20.0)	
Combination antifungal treatment	8 (2.2)	6 (1.8)	2 (5.7)	
No treatment	11 (3.0)	10 (3.0)	1 (2.9)	
Modification of antifungal agents	157 (43.0)	144 (43.6)	13 (37.1)	0.474
Duration of antifungal treatment (d), median (IQR)	17.0 (14.0–24.0)	17.5 (14.0–22.5)	17.0 (14.3–26.3)	0.561
Early removal of central venous catheter *	128 (35.1)	114 (34.5)	14 (40.0)	0.577
Appropriate antifungal treatment within 48 h	205 (56.2)	185 (56.1)	20 (57.1)	0.894
Treatment outcomes				
Responsiveness after effective antifungals ^&^				0.134
Within 72 h	136 (37.3)	124 (37.6)	12 (34.3)	
4–7 days	72 (19.7)	69 (20.9)	3 (8.6)	
More than 7 days	63 (17.3)	57 (17.3)	6 (17.1)	
Treatment failure	94 (25.7)	80 (24.4)	14 (40.0)	
Subsequent bacteremia	88 (24.1)	70 (21.2)	18 (51.4)	<0.001
Candidemia-attributable mortality	87 (23.8)	74 (22.4)	13 (37.2)	0.061

* Within the 3 days after onset of candidemia. ^&^ Responsiveness to antifungal agents was defined according to the consensus criteria of the Mycoses Study Group and European Organization for Research and Treatment of Cancer [20].

**Table 4 jof-08-01155-t004:** Univariate and multivariate logistic regression analysis for independent risk factors of candidemia-attributable mortality in pediatric patients with Candida bloodstream infection.

Variables	Univariate Analyses			Multivariate Regression Analysis		
	Odds Ratio	95% CI	*p* Value	Odds Ratio	95% CI	*p* Value
Patient age						
Neonates (<3 months old)	1.93	1.17–3.18	0.010			
Pediatric patients	1	(reference)				
Underlying chronic comorbidities						
No	1	(reference)		1	(reference)	
One	1.33	0.43–4.12	0.618	1.57	0.37–6.64	0.540
More than one chronic comorbidity	2.89	0.96–8.70	0.060	2.39	0.53–10.68	0.255
Septic shock at onset	11.16	5.55–28.76	<0.001	4.65	2.24–9.64	<0.001
Delayed CVC removal (>72 h)	4.08	2.44–6.84	<0.001	2.83	1.39–5.74	0.004
Subsequent bacteremia	1.27	0.74–2.20	0.386			
Breakthrough candidemia	4.69	2.48–8.87	<0.001	4.59	1.92–10.98	0.002
Delayed effective antifungal agents (>48 h)	2.02	1.21–3.37	0.007	1.12	0.50–2.52	0.792
Mixed *Candida*/bacterial bloodstream infections	2.04	0.98–4.25	0.056	1.52	0.53–4.32	0.436
Pathogens						
*Candida albicans*	1	(reference)				
*Non-albicans Candida* spp.	1.05	0.65–1.70	0.852			
Uncommon *Candida* spp.	1.04	0.52–2.09	0.908			
Case periods						
2005–2008	1.68	0.87–3.25	0.123			
2009–2012	1.23	0.60–2.53	0.576			
2013–2016	1.61	0.77–3.37	0.202			
2017–2020	1	(reference)				

## Data Availability

The datasets used/or analyzed during the current study are available from the corresponding author on reasonable request.

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
