# Peer review of "Pediatric Candida Bloodstream Infections Complicated with Mixed and Subsequent Bacteremia: The Clinical Characteristics and Impacts on Outcomes"

_jof, 2022, doi:10.3390/jof8111155_

Round 1

Reviewer 1 Report

Dear authors

1. Is the number of contaminated blood cultures determined?

2. Based on what evidence did you record the combined infection of candida and bacteria?

3. Are the samples that have been sent to the laboratory taken in a standard manner in terms of sampling criteria?

4. How do you reject the presence of normal flora in positive blood cultures?

Author Response

Dear reviewer:

      Thank you for your review and comments. Please see the attachment, thank you.

Best regard,

Tsai Ming Horng

Reviewer 2 Report

This study describes the clinical features, complications, risk factors, and outcomes associated with pediatric Candida/bacteria BSIs. The manuscript is written very well and it's worth publishing after minor corrections.

Minor corrections:

1.     Change "Candida" and "candida" with "Candida" throughout the manuscript.

2.     Page 4, Table 1: I suggest making the titles inside the table, such as "Patients demographics," and "Underlying Chronic Comorbidities, n (%)," .etc., bold so that they are more readable.

3. Page 4, Table 1: Use the same style when you mention "Total" here and throughout the manuscript.

4.     Page 4, section 3.1.: add the missing “%” after 40.5 in the “Gram-positive bacteria (n=17, 40.5)”.

5.     Page 5 & 6, Table 2 & 3: I could not see the footnotes?

6.     Page 5: Authors found that azole agent "In note, mixed Candida/bacterial BSIs were significantly more likely be treated with azole agents than monomicrobial Candida BSIs (57.1 vs. 37.5%, p = 0.014)". It would be better to describe in a few sentences the AST results and mention the number of resistant strains to azoles and/or other drugs if available.

Author Response

(The authors gave the same response as above.)

Round 2

Reviewer 1 Report

Dear editor

The manuscript is accept after minor revision.

The following comments should be edited.

1. Improve the manuscript in terms of grammar language.
2. The discussion section of the manuscript needs further expansion.
3. Use up-to-date articles in referencing.
4. The results section is not clear. Write in more detail.

Author Response

Dear reviewer:

     I appreciate your review and comments. Please see the attachement. However, for writing the result section more clear (the forth comments), would you please be more specific, which part of the result section should I revise to make it more clear? I appreciarte your response and review, thank you.

Best regard,

Tsai Ming Horng
